# Prostacyclin Analogues Inhibit Platelet Reactivity, Extracellular Vesicle Release and Thrombus Formation in Patients with Pulmonary Arterial Hypertension

**DOI:** 10.3390/jcm10051024

**Published:** 2021-03-02

**Authors:** Aleksandra Gąsecka, Marta Banaszkiewicz, Rienk Nieuwland, Edwin van der Pol, Najat Hajji, Hubert Mutwil, Sylwester Rogula, Wiktoria Rutkowska, Kinga Pluta, Ceren Eyileten, Marek Postuła, Szymon Darocha, Zenon Huczek, Grzegorz Opolski, Krzysztof J. Filipiak, Adam Torbicki, Marcin Kurzyna

**Affiliations:** 11st Chair and Department of Cardiology, Medical University of Warsaw, 02-097 Warsaw, Poland; hubert.mutwil@gmail.com (H.M.); sylwesterrogula@o2.pl (S.R.); wiktoria.rutkowska97@gmail.com (W.R.); plutakinga.01@gmail.com (K.P.); zhuczek@wp.pl (Z.H.); grzegorz.opolski@wum.edu.pl (G.O.); krzysztof.filipiak@wum.edu.pl (K.J.F.); 2Laboratory of Experimental Clinical Chemistry and Vesicle Observation Centre, Amsterdam University Medical Centre, University of Amsterdam, 1105 AZ Amsterdam, The Netherlands; r.nieuwland@amsterdamumc.nl (R.N.); e.vanderpol@amsterdamumc.nl (E.v.d.P.); n.hajji@amsterdamumc.nl (N.H.); 3Department of Pulmonary Circulation, Thromboembolic Diseases and Cardiology, Centre of Postgraduate Medical Education, European Health Centre Otwock, 05-400 Otwock, Poland; marta.banaszkiewicz@gmail.com (M.B.); szymon.darocha@ecz-otwock.pl (S.D.); atorbicki@gmail.com (A.T.); marcin.kurzyna@ecz-otwock.pl (M.K.); 4Biomedical Engineering and Physics, Amsterdam University Medical Centre, University of Amsterdam, 1105 AZ Amsterdam, The Netherlands; 5Department of Experimental and Clinical Pharmacology, Centre for Preclinical Research and Technology, Medical University of Warsaw, 02-091 Warsaw, Poland; cereneyileten@gmail.com (C.E.); mpostula@wum.edu.pl (M.P.)

**Keywords:** extracellular vesicles, platelet reactivity, prostacyclin analogues, pulmonary arterial hypertension, thrombus formation

## Abstract

(1) Background: Prostacyclin analogues (epoprostenol, treprostinil, and iloprost) induce vasodilation in pulmonary arterial hypertension (PAH) but also inhibit platelet function. (2) Objectives: We assessed platelet function in PAH patients treated with prostacyclin analogues and not receiving prostacyclin analogues. (3) Methods: Venous blood was collected from 42 patients treated with prostacyclin analogues (49.5 ± 15.9 years, 81% female) and 38 patients not receiving prostacyclin analogues (55.5 ± 15.6 years, 74% female). Platelet reactivity was analyzed by impedance aggregometry using arachidonic acid (AA; 0.5 mM), adenosine diphosphate (ADP; 6.5 µM), and thrombin receptor-activating peptide (TRAP; 32 µM) as agonists. In a subset of patients, concentrations of extracellular vesicles (EVs) from all platelets (CD61^+^), activated platelets (CD61^+^/CD62P^+^), leukocytes (CD45^+^), and endothelial cells (CD146^+^) were analyzed by flow cytometry. Platelet-rich thrombus formation was measured using a whole blood perfusion system. (4) Results: Compared to controls, PAH patients treated with prostacyclin analogues had lower platelet reactivity in response to AA and ADP (*p* = 0.01 for both), lower concentrations of platelet and leukocyte EVs (*p* ≤ 0.04), delayed thrombus formation (*p* ≤ 0.003), and decreased thrombus size (*p* = 0.008). Epoprostenol did not affect platelet reactivity but decreased the concentrations of platelet and leukocyte EVs (*p* ≤ 0.04). Treprostinil decreased platelet reactivity in response to AA and ADP (*p* ≤ 0.02) but had no effect on the concentrations of EVs. All prostacyclin analogues delayed thrombus formation and decreased thrombus size (*p* ≤ 0.04). (5) Conclusions: PAH patients treated with prostacyclin analogues had impaired platelet reactivity, EV release, and thrombus formation, compared to patients not receiving prostacyclin analogues.

## 1. Introduction

Pulmonary arterial hypertension (PAH) is a devastating and incurable disease leading to right-heart failure and premature death [1]. In patients with PAH, elevated pressure in the pulmonary arteries is a consequence of increased concentrations of the vasoconstricting substances (thromboxane A2, endothelin 1) and decreased concentrations of vasodilating substances (nitric oxide, prostacyclin) [2]. This elevated ratio between vasoconstricting and vasodilating mediators leads to remodeling of the pulmonary arterioles, increased contractility of the arterioles, and thrombus formation in the lumen [2]. Increased shear stress in the pulmonary circulation activates platelets, which play a crucial role in the development and progression of PAH through the secretion of vasoconstricting, proinflammatory, and prothrombotic mediators, thereby promoting further remodeling [3]. Activated platelets also release platelet-derived extracellular vesicles (EVs), which mediate the proinflammatory and prothrombotic effects of activated platelets in PAH [4,5]. Elevated concentrations of platelet EVs were observed in many clinical conditions associated with platelet activation including PAH regardless of PAH etiology, including idiopathic and genetically determined PAH [6,7]. Activated platelets and platelet EVs form complexes with leukocytes, leading to leukocyte activation via P-selectin (CD62P), the release of proinflammatory cytokines and leukocyte EVs, and the production of tissue factor, the initiator of clotting activator in vivo, by leukocytes [8]. Platelet EVs and leukocyte EVs promote damaging of the pulmonary endothelium and remodeling of the pulmonary arterial smooth muscle cells, thereby contributing to PAH progression [9,10]. Patients with PAH also have increased concentrations of EVs from endothelial cells, which correlate to PAH hemodynamic severity and are associated with adverse clinical events [11,12]. Altogether, EVs seem to modulate the development and progression of PAH, with their role depending on the EV cellular origin, molecular cargo, and exposed molecules.

According to the guidelines of the European Society of Cardiology, the currently available options to treat PAH target the three different pathways, which underlie PAH: (i) Increased concentration of endothelin-1, (ii) decreased concentration of nitric oxide, and (iii) decreased concentration of prostacyclin [13]. Consequently, the available pharmacotherapy includes: (i) Endothelin receptor antagonists (ERA; ambrisentan, bosentan, macitentan), which decrease the vasoconstricting effects of endothelin-1; (ii) phosphodiesterase type 5 inhibitors (PDE-5i; sildenafil, tadalafil) and soluble guanyl cyclase stimulators (riociguat), which increase the concentration of nitric oxide; and (iii) prostacyclin analogues (epoprostenol, treprostinil, and iloprost), which enhance the concentration and vasodilating effects of prostacyclin [13].

Prostacyclin analogues induce vasodilation in PAH and inhibit platelets, potentially increasing patients’ bleeding risk. However, the antiplatelet effects of different prostacyclin analogues have never been compared head to head. We hypothesized that (i) patients treated with prostacyclin analogues have lower platelet reactivity compared to patients not receiving prostacyclin analogues, and (ii) the antiplatelet effects of prostacyclin analogues (epoprostenol, treprostinil, and iloprost) may vary. Accordingly, the goal of the PAPAYA (Platelet Reactivity and Treatment With Prostacyclin Analogues in Pulmonary Arterial Hypertension) trial was (i) to compare platelet function, defined as platelet reactivity, platelet EVs concentration, and thrombus formation, in patients with PAH treated with prostacyclin analogues on top of ERA and/or PDE5i and patients treated only with ERA and/or PDE5i; and (ii) to compare the antiplatelet effect of different prostacyclin analogues.

## 2. Methods

### 2.1. Study Design

PAPAYA was an investigator-initiated, prospective, observational study conducted at the Department of Pulmonary Circulation, Thromboembolic Diseases and Cardiology, Centre of Postgraduate Education Medical, European Health Centre Otwock, Poland in collaboration with the First Chair and Department of Cardiology, Medical University of Warsaw, Poland and the Vesicle Observation Centre, Amsterdam University Medical Centres (UMC), The Netherlands. The study protocol was designed in compliance with the Declaration of Helsinki and approved by the Medical University of Warsaw Ethics Committee (approval number: KB/138/217). The protocol was also registered in the ClinicalTrials database (NCT04578223).

### 2.2. Selection of Participants

Study inclusion and exclusion criteria are listed in Table 1. Patients were eligible for enrolment if they (i) had PAH diagnosed according to the criteria of the European Society of Cardiology and European Respiratory Society [13], (ii) had PAH confirmed during right-heart catheterization [14], (iii) were treated with prostacyclin analogues on top of ERA and PDE-5i (study group) or were only treated with ERA and PDE-5i (control group) for at least 1 month, and (iv) provided written informed consent to participate in the study. Exclusion criteria were known coagulopathy, active pathological bleeding or a history of bleeding disorder, severe thrombocytopenia (platelet count < 50,000/μL), severe chronic renal failure (estimated glomerular filtration rate < 30 mL/min), or severe liver insufficiency (Child-Pugh class C) and antiplatelet therapy with acetylsalicylic acid or P2Y12 antagonists.

### 2.3. Trial Schedule and Blinding

The trial schedule is presented in Figure 1A. Patients were recruited for the study by independent operators (MB, SD) who were not involved in the sample analysis. Blood collection was performed at a single timepoint. All samples were measured and analyzed by operators blinded to clinical data (NH, EvdP). Statistical analysis was performed by another operator (AG). The double blinding was not applied due to the differences in the administration routes of prostacyclin analogues, ERA and PDE-5i.

### 2.4. Study Treatment

In a study group, epoprostenol was administered intravenously through a surgically placed central venous catheter equipped with a portable pump at doses ranging from 27 ng/kg/min to 117 ng/kg/min (median: 41 ng/kg/min). Treprostinil was administered subcutaneously by continuous infusion using an infusion set connected to an infusion pump at doses ranging from 12 ng/kg/min to 189 ng/kg/min (median: 42.5 ng/kg/min). Six patients were implanted with the Lenus Pro pump [15] and received continuous intravenous infusion of treprostinil at doses ranging from 37 ng/kg/min to 86 ng/kg/min (median: 59 ng/kg/min). Iloprost was delivered by a dedicated nebulizer (II-ne AAD or Breelib) from 6 to 9 inhalations daily, corresponding to 15–45 micrograms of the iloprost at mouthpiece [16]. In both groups, ERA and PDE-5i were administered orally at the discretion of the treating physician.

All patients with a positive response to the acute vasoreactivity test were administered calcium channel blockers at high doses. Patients with the symptoms of right-heart failure and fluid retention were administered diuretics at the discretion of the treating physician. Continuous oxygen therapy was a complementary therapy in patients with constant hypoxemia <8 kPa (60 mmHg) according to the guidelines [13]. Finally, all patients received standard treatment depending on the individual clinical characteristics and comorbidities, including oral anticoagulation, β-blockers, and statins.

### 2.5. Clinical Data Collection

The following data were collected at baseline: Demographics (age, gender), body mass index, PAH etiology and World Health Organization (WHO) class, cardiovascular risk factors (smoking, hypertension, dyslipidemia, diabetes), and history of cardiovascular disease (myocardial infarction, stroke). In addition, routine laboratory parameters and pharmacotherapy were recorded. Data regarding thrombotic and bleeding events since the index hospitalization were recorded up to 9 months after the end of the recruitment.

### 2.6. Samples Collection and Handling

Blood collection was performed in the afternoon (between 14:00 and 16:00) from non-fasting patients. In case of iloprost, blood collection was performed within 30 min after inhalation of the last dose. Samples were collected and processed by experienced operators (S.R., H.M., W.R.), according to the guidelines to study platelet reactivity and plasma EV concentrations [17]. Briefly, blood was collected in two 1.6 mL hirudin plastic tubes and one 8.2 mL 0.109 M citrated (S-Monovette, Sarstedt, Nümbrecht, Germany) via antecubital vein puncture using a 19-gauge needle, without tourniquet. The first 2 mL were discarded to avoid platelet pre-activation. Platelet reactivity and platelet-rich thrombus formation were evaluated simultaneously by 2 operators using impedance aggregometry and whole blood perfusion system, respectively, at the First Chair and Department of Cardiology, Medical University of Warsaw, Poland. Both measurements were done within 5 min after blood collection due to the short half-life time of prostacyclin analogues (epoprostenol: 6 min, iloprost: 20–30 min, treprostinil: 1.5 h). Platelet-depleted plasma for EV analysis was prepared using double centrifugation using the following parameters: (i) Centrifugation speed: 2.500 g, (ii) time: 15 min, (iii) temperature: 20 °C, and (iv) acceleration speed: 1, no brake [18]. The first centrifugation step was done in the whole blood collection tubes and about 3.5 mL plasma was collected 10 mm above the buffy coat. The second centrifugation step was done in polypropylene centrifuge tubes (Greiner Bio-One B.V, Alphen aan den Rijn, Netherlands). Supernatant (platelet-depleted plasma) was collected 5 mm above the bottom, transferred into separate tubes (Greiner Bio-One B.V), and mixed by pipetting. Finally, plasma was transferred to 1.5 mL low-protein binding Eppendorf tubes (Thermo Fisher Scientific, Nieuwegein, Netherlands) and stored in −80 °C until analyzed. EV analysis was performed in the Vesicle Observation Centre, Amsterdam University Medical Centre, The Netherlands. Prior to analysis, samples were thawed for 1 min in a water bath (37 °C).

### 2.7. Endpoints

The primary endpoint was the difference in platelet reactivity between patients treated with prostacyclin analogues and treated with ERA and/or PDE5-i. The secondary endpoints were differences in (i) the concentration of EVs from platelets, leukocytes, and endothelial cells; and (ii) platelet-rich thrombus formation parameters between patients treated with prostacyclin analogues and treated with only ERA and/or PDE5i. The tertiary endpoints were the differences in the abovementioned platelet function parameters between patients treated with different prostacyclin analogues (epoprostenol, treprostinil, and iloprost). The study was not powered for mortality or other adverse events.

### 2.8. Laboratory Assays

Since impedance aggregometry was available during the whole study period, platelet reactivity was measured in all patients in the study group (*n* = 42) and control group (*n* = 38). The infrastructure required for blood handling for EV analysis and whole blood perfusion system became available after the study started in March 2019. Therefore, both the concentrations of EVs and thrombus formation were measured in a subset of patients in the study and control groups (both *n* = 20).

#### 2.8.1. Platelet Reactivity

Platelet reactivity was assessed using multiple electrode aggregometry (Roche Diagnostics, Mannheim, Germany). Arachidonic acid (AA, 0.5 mM: ASPI test), adenosine diphosphate (ADP, 6.5 µM: ADP test), and thrombin receptor-activating peptide-6 (TRAP, 32 µM: TRAP test) were used to activate platelets. Unstimulated whole blood served as a negative control [19].

#### 2.8.2. Flow Cytometry

Flow cytometry (A60-Micro, Apogee Flow Systems, Hertfordshire, UK) was used to determine the concentration of EV subtypes in platelet-depleted plasma. We diluted samples in phosphate-buffered saline (PBS) to a count rate below 3000 events/s to prevent swarm detection. Diluted samples were measured using the following settings: (i) Time: 120 s, (ii) flow rate: 3.01 μL/min, and (iii) trigger threshold: 14 arbitrary units of the side scatter detector, corresponding to a side scattering cross section of 10 nm^2^. By concentrations of EVs per mL plasma, we mean particles (i) exceeding the side scatter threshold, (ii) with a diameter >200 nm according to applied flow cytometry scatter ratio (Flow-SR) [20], (iii) with a refractive index <1.42 to omit nonspecifically labelled chylomicrons [21], and (iv) positive at the fluorescence detector(s) for the used label(s). Due to the detection limit of our flow cytometer, the EVs measured in this study predominantly correspond to microparticles, not to exosomes. We aimed to label EVs released by all platelets and megakaryocytes (CD61^+^), activated platelets (CD62P^+^), leukocytes (CD45^+^), and endothelial cells (CD146^+^). To improve the reproducibility of the flow cytometry analysis, we (i) reported our results according to the standardized framework (MIFlowCyt-EV) [22], (ii) calibrated all detectors, (iii) applied Flow-SR to determine the diameter and refractive index of EVs [20], and (iv) automated data calibration and processing with a custom-built software [23]. All relevant details about sample preparation, assay controls, instrument calibration, data acquisition, and EV characterization can be found in Appendix A.

#### 2.8.3. Whole Blood Perfusion System

Platelet-rich thrombus formation was analyzed on the T-TAS instrument (Zacros, Fujimori Kogyo Co. Ltd., Tokyo, Japan) using the PL microchip containing capillary channels coated with type 1 collagen under arterial shear rate (2000 s^−1^) [24]. Briefly, hirudin anticoagulated whole blood (320 µL) was pipetted in the reservoir and perfused at 37 °C through the PL chip by a pneumatic pump. The process of thrombus formation was monitored by flow pressure changes in the capillary using the pressure sensor located between the pump and the reservoir. As thrombus formation proceeded on the coated surface, the capillary was gradually occluded, increasing the flow pressure. Based on the flow pressure pattern, the following 3 parameters are used to analyze the results: (i) T10, representing the duration for the flow pressure to increase from baseline to 10 kPa due to partial occlusion of microcapillaries, defined as the onset of thrombus formation; (ii) occlusion time, coinciding with a pressure of 60 kPa, defined as the complete occlusion of the capillary; and (iii) area under curve (AUC), which is an area under the flow pressure curve (under 60 kPa) for 12 min after the start of assay, used to quantify the size of platelet-rich thrombus [24].

### 2.9. Statistical Analysis

Because there were insufficient data to assess the impact of prostacyclin analogues on platelet reactivity, concentrations of EVs, and thrombus formation in patients with PAH at the time of study initiation, the standard deviation (SD) and mean difference between the groups were estimated based on our previous in vitro experiments [19]. The required sample size for the primary endpoint (platelet reactivity) was calculated by a 2-sided *t*-test at a significance level of 0.05, assuming (i) SD in each group ± 1.5, (ii) mean difference between the groups = 1, and (iii) nominal test power = 0.9. Based on this estimation, we used a sample size of 37 patients per group to observe significant differences in platelet function parameters. The required sample size for the secondary endpoint (concentration of EVs and platelet-rich thrombus formation parameters) was calculated by a 2-sided *t*-test at a significance level of 0.05 with the following assumptions: (i) SD in each group ± 1.0, (ii) mean difference between the groups = 1, and (iii) nominal test power = 0.8. Based on this estimation, we used a sample size of 17 patients per group to observe significant differences in secondary endpoints. Altogether, 80 patients were included in the study.

Statistical analysis was conducted using IBM SPSS Statistics, version 24.0 (IBM, New York, NY, USA). Categorical variables were presented as number and percent and compared using Fisher’s exact test. The Shapiro–Wilk test was applied to check the distribution of continuous variables. Continuous variables were presented as mean and SD or median with interquartile range. The Mann–Whitney U test was used to compare the values between the 2 treatment arms (prostacyclin analogues versus ERA/PDE-5i). The Kruskal–Wallis test with Dunn’s correction for multiple comparisons was used to compare the values between the subgroups treated with different prostacyclin analogues (epoprostenol, treprostinil). Iloprost was excluded from the subanalysis due to unrepresentative sample size. Correlations were analyzed using the Spearman correlation coefficient test. Mortality and other adverse events were reported descriptively. A *p*-value below 0.05 was significant.

## 3. Results

### 3.1. Study Population

Patients’ flow diagram is shown in Figure 1B. Of the 128 patients who were treated with prostacyclin analogues or ERA/PDE5i in the Department of Pulmonary Circulation, Thromboembolic Diseases and Cardiology, European Health Centre Otwock, Poland between July 2017 and November 2019, 80 patients were included in the study, 42 patients in the study group, and 38 patients in the control group. Among the patients treated with prostacyclin analogues, 30 patients were treated with treprostinil, 8 with epoprostenol and 4 with iloprost. Patient baseline characteristics are presented in Table 2. In the study group, the idiopathic PAH (iPAH) was slightly more common than in the control group (36% vs. 25%, *p* = 0.04), but there were no differences in PAH functional class. Since patients with PAH associated with connective tissue disease (CTD) tend to have more activated platelets compared to iPAH [25], both groups contained comparable numbers of patients with CTD. Cardiovascular risk factors, comorbidities, and laboratory characteristics were comparable in both groups. Although prostacyclin analogues increase the risk of thrombocytopenia, the median platelet count was comparable in both groups (189 ± 48 × 10^3^ per μL in the study group and 196 ± 55 × 10^3^ per μL in the control group). In both groups, ~50% of patients were treated with ERA and ~90% of patients were treated with PDE-5i. Oral anticoagulants were administered to ~20% of patients. Since both ERA and/or PDE5i and oral anticoagulants affect platelet function and thrombus formation parameters [26,27,28], the pharmacotherapy was well balanced between the groups, minimizing the risk of potential bias.

### 3.2. Platelet Function

Figure 2 shows the platelet reactivity in patients treated with prostacyclin analogues and control patients, including resting platelets (not stimulated with any agonist; negative control) and platelets stimulated with AA (ASPI test), ADP (ADP test), and TRAP (TRAP test). Patients treated with prostacyclin analogues responded less to AA and ADP compared to control patients (Figure 2A, *p* = 0.01 for both). Whereas epoprostenol did not affect platelet reactivity in vitro, treprostinil decreased platelet reactivity in ASPI test and ADP test (Figure 2B, *p* ≤ 0.02 for both).

Figure 3 shows the concentrations of EVs measured in platelet-depleted plasma by flow cytometry. Patients treated with prostacyclin analogues had comparable concentrations of EVs from platelets (CD61^+^), but lower concentrations of EVs from activated platelets (CD62P^+^) and leukocytes, compared to control patients (*p* = 0.04 and *p* = 0.01, respectively). There was also a trend toward a lower concentration of EVs from endothelial cells (*p* = 0.08). Whereas patients treated with treprostinil had comparable concentrations of EVs as control group, patients treated with epoprostenol had lower concentrations of platelet EVs (CD61^+^ and CD62P^+^, *p* ≤ 0.04 for both) and leukocyte EVs (*p* = 0.04).

Figure 4 shows the platelet-rich thrombus formation measured in a whole blood perfusion system. In patients treated with prostacyclin analogues, clotting initiation (T10) and time to occlusion of the capillaries were delayed (*p* = 0.002 and *p* = 0.003, respectively), and thrombus size (AUC) was decreased (*p* = 0.008) compared to control patients. Both epoprostenol and treprostinil delayed thrombus formation and decreased thrombus size (*p* ≤ 0.04).

In 5 out of 20 patients treated with prostacyclin analogues (25%), no thrombus was formed, i.e., total occlusion of the capillaries was not achieved during the measurement time in 4 patients on treprostinil and 1 on iloprost. In one patient on treprostinil, the thrombus formation was not even initiated (T10 was not achieved). In contrast, a thrombus was formed in all patients from the control group and all patients treated with epoprostenol. Examples of pictures obtained during thrombus formation in the whole blood perfusion system are shown in Figure 5, including (i) a control patient who achieved total occlusion of the microchip after 3 min 40 s (Figure 5A), (ii) a patient treated with epoprostenol who achieved clotting after 9 min 39 s (Figure 5B), and (iii) a patient treated with treprostinil who did not achieve clotting during the measurement time of 12 min (Figure 5C). The full movies are attached in the Appendix A.

We did not find any correlations between the dose of treprostinil and any of the evaluated platelet function parameters (Appendix A). Due to the higher prevalence but lower severity of PAH in female patients, we compared all measured platelet function parameters in female and male patients. We did not find any gender-associated differences except for higher concentrations of EVs from platelets (CD61+) and leukocytes (CD45+) in females (*p* ≤ 0.03 for both). To further evaluate the effect of potential clinical confounders on CD61+ and CD45+ EVs, we performed a multivariate regression analysis, taking into account the concentrations of CD61+ and CD45+ EVs above the median as a dependent variable and (i) age, (ii) gender (female), (iii) PAH severity (WHO class), and (iv) therapy with prostacyclin analogues as independent variables. We found that both female gender and therapy with prostacyclin analogues were independently associated with the concentrations of CD61+ and CD45+ EVs above the median. We attached the data to the Appendix A.

### 3.3. Clinical Outcomes

There were no deaths and no thrombotic events during the study up to 9 months after the recruitment finished. There were two major bleeding events on treprostinil, i.e., hematomas at the site of implantation of the infusion pump, which required blood transfusion and reoperation in one case and conservative treatment in the second case. In both patients, no thrombus formation was observed in the whole blood perfusion system, and in one case thrombus formation (T10), was not even initiated (please see Appendix A).

### 3.4. Compliance

Since all patients were regularly admitted to the hospital every 3–6 months within the therapeutic program, compliance was checked during the visits using the infusion pump reader (treprostinil, epoprostenol) or the inhalator reader (iloprost), indicating the amount of the drug administered prior to the visit. All patients were compliant before inclusion in the study and blood collection.

## 4. Discussion

PAPAYA is the first clinical study investigating the effect of prostacyclin analogues on multiple parameters of platelet function, namely platelet reactivity, EV release, and thrombus formation under flow conditions in whole blood. Moreover, we attempted to compare the effects of prostacyclin analogues on platelet function head to head in an investigator-blinded way. Finally, PAPAYA applied the recently standardized guidelines to study EV, including determination of the EV diameter and refractive index by Flow-SR [20], calibrated flow cytometry, and dedicated software to automate data processing [29].

The main finding of our study is that patients with PAH treated with prostacyclin analogues have impaired platelet reactivity, decreased platelet and leukocyte EV release, and impaired thrombus formation compared to patients not receiving prostacyclin analogues. These results may have both negative and positive clinical implications.

On the one hand, the antiplatelet effect of prostacyclin and its analogues may increase patients’ bleeding risk. Preliminary evidence regarding the antiplatelet role of prostacyclin in animal models dates back to 30 years ago, when it was shown that prostacyclin inhibits platelet adhesion and thrombus formation on the subendothelium [30] and prevents thrombus formation in microcirculation [31]. More recently, prostacyclin was shown to reverse platelet stress fiber formation, causing platelet aggregate instability in vitro [32]. The mechanisms underlying the antiplatelet effects of prostacyclin and other prostanoids on platelet function have been summarized in recent excellent reviews [33,34]. Still, the hitherto published data are based on in vitro experiments [35,36], studies in healthy volunteers [37], and small (*n* < 30) observational studies in pediatric patients with PAH [38,39]. In line with previous studies, we showed that prostacyclin analogues inhibit platelet reactivity and decrease platelet EV release. Additionally, we showed that impaired platelet function is associated with impaired thrombus formation, explaining the pathophysiological mechanisms underlying increased risk of bleeding in patients treated with prostacyclin analogues [40,41]. We observed that two patients who did not achieve clotting in the whole blood perfusion systems subsequently experienced bleeding. These observations imply the potential application of the Total-Thrombus Analysis System to predict bleeding events in patients with PAH if confirmed in future trials.

On the other hand, since EVs modulate the development and progression of PAH [9,10,11,12], the decreased release of proinflammatory and prothrombotic EVs from platelets and leukocytes by prostacyclin analogues may slow the progression of PAH. Our observation allows us to speculate that the modulation of EV concentration might be one of the mechanisms underlying the beneficial effect of prostacyclin analogues in PAH. The “anti-EV” effect has also been observed for the P2Y12 antagonists, which block platelets by the same intracellular pathway as prostacyclin analogues—increasing the intracellular concentration of cyclin adenosine monophosphate (cAMP) [42,43].

Finally, we showed that the mechanisms underlying the antiplatelet effect of prostacyclin analogues differ, although all prostacyclin analogues delay thrombus formation and limit thrombus size. There are four types of prostacyclin receptors on platelets: IP and DP, which impair aggregation, and TP and EP3, which promote aggregation [44]. In our study, treprostinil, which binds to IP and DP receptors only, decreased platelet reactivity in response to AA and ADP. IP and DP receptors are both coupled to the Gs-protein that activates adenylyl cyclase and stimulates cAMP formation. High concentrations of cAMP make platelets resistant to activation by any agonist [45]. Although we did not find any effect of treprostinil on EV release, other authors showed that treprostinil decreased the concentrations of platelet EVs in pediatric PAH, with no effect on leukocyte and endothelial EVs [39]. It is likely that different populations and small sample sizes account for these differences.

In contrast to previous findings [46], in our study, epoprostenol had no significant effect on platelet reactivity but decreased the concentrations of platelet and leukocyte EVs. Since epoprostenol binds both to the anti-aggregatory IP and DP receptors and to the pro-aggregatory EP3 receptor, the effect of epoprostenol on platelet reactivity might be less potent [47]. The mechanism underlying the “anti-EV” effect of epoprostenol, in turn, could be associated with the fact that epoprostenol decreased the thrombus size more than treprostinil (Figure 4D). Since platelet EVs are released after reversible platelet-rich thrombus is formed [48], by decreasing the thrombus size, epoprostenol might decrease EV release as well. In line with our results, in a recent cross-sectional study comprising 73 patients with PAH, a negative correlation between the dose of epoprostenol and the concentration of platelet EVs was observed [7].

Altogether, the differences in the antiplatelet effects of prostacyclin analogues might be explained by their various potency and affinity to bind prostacyclin receptors. Because our study was not adequately powered to compare different prostacyclin analogues head to head (tertiary endpoints), our results are preliminary and require confirmation in future trials.

## 5. Limitations

The main limitation of our study is the nonrandomized study design. For example, despite the lack of statistically significant differences, the largest percentage of patients in the study group had WHO class III PAH (50%), and the largest percentage of patients in the control group had WHO class II PAH (55%). Concurrently, the NT-proBNP level was considerably higher in the study group, indicating more advanced right ventricular dilatation and impaired systolic function. Therefore, despite the differences in the primary and secondary endpoints between the groups, the results should be interpreted with caution. Second, patients were assessed when already on treatment, which did not allow us to evaluate the effect of prostacyclin analogues initiation on platelet function. Third, since combined therapy including ERA, PDE5i, and prostacyclin analogues is recommended in WHO functional class II-IV PAH, such therapy was also used it our study, which did not allow us to compare the effects of monotherapy with prostacyclin analogues in the study group vs. no therapy in the control group. Fourth, although this study was larger than the previous studies investigating the antiplatelet effects of prostacyclin analogues, the patient number in the various prostacyclin groups were low, hampering the head-to-head comparison of these drugs. Moreover, our patients were not screened for the presence of a bone morphogenetic protein receptor type II (BMPR2) mutation. Inactivating mutation in the BMPR2 gene in patients with PAH has been linked to vascular smooth muscle cell proliferation and upregulation of platelet adhesion. Therefore, it cannot be excluded that the mutation status has potential influences on the parameters of platelet function evaluated in this study. Finally, lack of clinical endpoints makes the results hypothesis-generating rather than ultimately proving that impaired platelet function is associated with bleeding in PAH.

## 6. Conclusions

We found that patients with PAH who are treated with prostacyclin analogues may have increased bleeding risk due to both impaired platelet function and thrombus formation compared to patients not receiving prostacyclin analogues. On the other hand, the decreased release of proinflammatory and prothrombotic EVs from platelets and leukocytes by prostacyclin analogues may slow the progression of PAH. Further randomized clinical studies are required to compare different prostacyclin analogues head to head and to determine the optimal treatment regimen in patients with increased risk of thrombosis or bleeding.

## Figures and Tables

**Figure 1 jcm-10-01024-f001:**
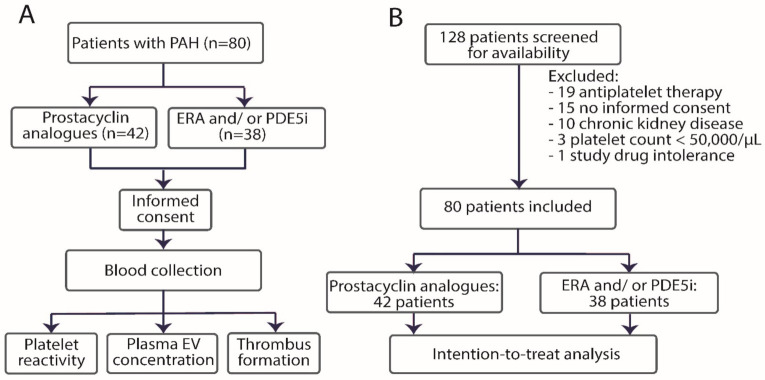
Trial schedule (**A**) and patients’ flow diagram (**B**). PAH—pulmonary arterial hypertension, ERA—endothelin receptor antagonists, PGE-5i—phosphodiesterase-5 inhibitors.

**Figure 2 jcm-10-01024-f002:**
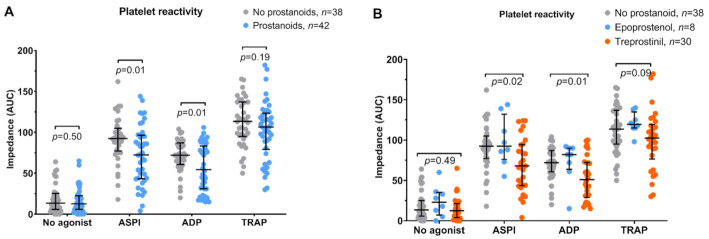
(**A**) Platelet reactivity in blood in response to arachidonic acid (ASPI test), adenosine diphosphate (ADP test), and thrombin receptor-activating peptide-6 (TRAP test). Unstimulated platelets (no agonist) were used as a negative control. Panel A: Patients receiving prostacyclin analogues and not treated with prostacyclin analogues. Values were compared using the Mann–Whitney U test and shown as the median and interquartile range. (**B**) Comparison between no prostacyclin analogues, epoprostenol, and treprostinil). Values were compared using the Kruskal—Wallis test with Dunn’s correction for multiple comparisons and shown as the median and interquartile range. AUC—area under the curve.

**Figure 3 jcm-10-01024-f003:**
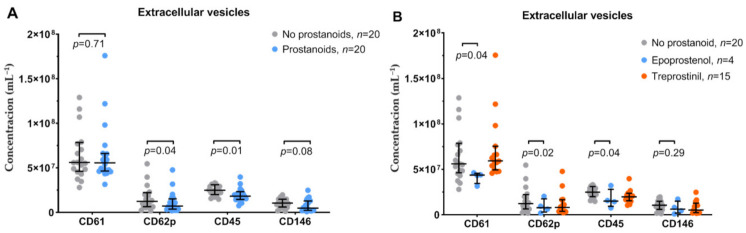
Concentrations of extracellular vesicles (EVs) from platelets (CD61, CD62P), leukocytes (CD45), and endothelial cells (CD146) measured with flow cytometry in platelet-depleted plasma. (**A**) Patients receiving prostacyclin analogues and not treated with prostacyclin analogues. Values were compared using the Mann–Whitney U test and shown as the median and interquartile range. (**B**) Comparison between no prostacyclin analogues, epoprostenol, and treprostinil. Values were compared using the Kruskal–Wallis test with Dunn’s correction for multiple comparisons and shown as the median and interquartile range.

**Figure 4 jcm-10-01024-f004:**
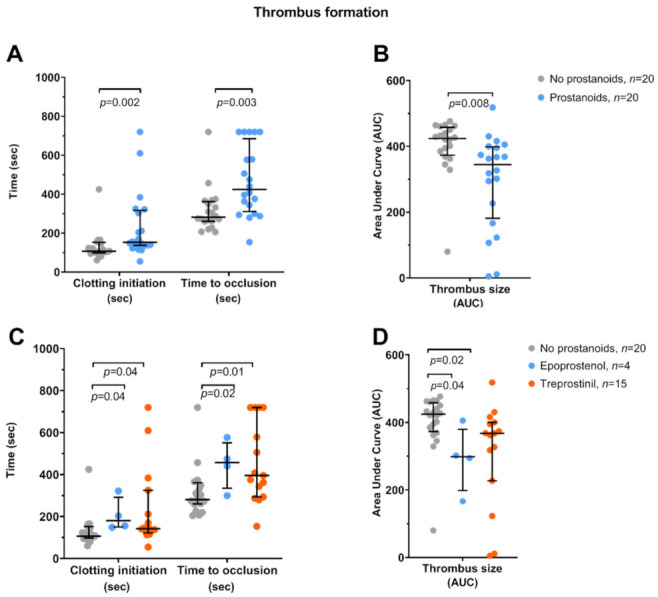
Platelet-rich thrombus formation measured in a whole blood perfusion system using the microchip containing capillaries coated with type 1 collagen under arterial shear rate (2000 s^−1^). (**A**,**B**) Patients receiving prostacyclin analogues and not treated with prostacyclin analogues. Values were compared using the Mann–Whitney U test and shown as the median and interquartile range. (**C**,**D**) Comparison between no prostacyclin analogues, epoprostenol, and treprostinil. Values were compared using the Kruskal–Wallis test with Dunn’s correction for multiple comparisons and shown as the median and interquartile range. T10—onset of thrombus formation. Occlusion time—complete occlusion of the capillary. Area under curve (AUC)—parameter reflecting platelet-rich thrombus size.

**Figure 5 jcm-10-01024-f005:**
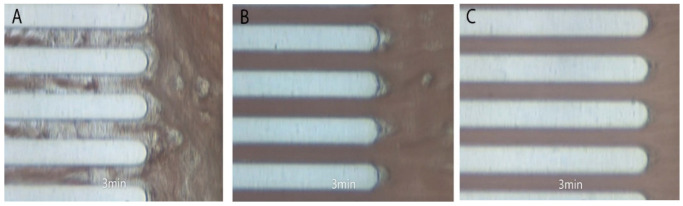
Examples of pictures obtained during thrombus formation in the whole blood perfusion scheme. (**A**) (i) A control patient who achieved total occlusion of the microchip after 3 min 40 s, (**B**) (ii) a patient treated with epoprostenol who achieved clotting after 9 min 39 s, and (**C**) (iii) a patient treated with treprostinil who did not achieve clotting during the measurement time of 12 min. All pictures were made at 3 min. The full movies are attached in the online Appendix A.

**Table 1 jcm-10-01024-t001:** Study inclusion and exclusion criteria. eGFR—estimated glomerular filtration rate.

Inclusion Criteria	Exclusion Criteria
Age > 18 years Informed consent to participate in the study Pulmonary arterial hypertension confirmed with right heart catheterization Treatment with prostacyclin analogues (epoprostenol, treprostinil, iloprost)—study group Treatment with endothelin receptor antagonists and phosphodiesterase type 5 inhibitors—control group	Known coagulopathy Active pathological bleeding Known history of bleeding disorder Severe thrombocytopenia (platelet count < 50,000/μL) Need for antiplatelet therapy with acetylsalicylic acid or P2Y12 antagonists Severe chronic renal failure (eGFR < 30 mL/min) Severe liver insufficiency (Child-Pugh class C) Known pregnancy, breast-feeding, or intention to become pregnant during the study period Study drug intolerance Participation in any previous study with prostacyclin analogues

**Table 2 jcm-10-01024-t002:** Patient characteristics.

	Study Group (*n* = 42)		Control Group (*n* = 38)		
Parameter	Number (%) or Mean ± SD or Median (IQR)		Number (%) or Mean ± SD or Median (IQR)		*p*-Value
Age, years—mean ± SD	49.5	15.9	55.5	15.7	0.11
Female gender—number (%)	34	81	28	74	0.27
BMI—mean ± SD	26.4	6.2	26.9	4.7	0.84
PAH etiology—number (%)					
Congenital heart disease	3	7	7	18	0.18
Connective tissue disease	3	7	6	16	0.29
Idiopathic	36	86	25	66	0.04
WHO functional class—number (%)					
II	18	43	21	55	0.38
III	21	50	15	40	0.38
IV	3	7	2	5	1.00
Comorbidities—number (%)					
Arterial hypertension	17	45	19	50	0.50
Diabetes mellitus	6	16	7	18	0.76
Dyslipidemia	10	26	9	24	1.00
Smoking	2	5	3	8	0.66
History of CVD—number (%)					
Myocardial infarction	3	8	5	13	0.47
Stroke	3	8	2	5	1.00
Laboratory parameters					
ALT, U/L—median (IQR)	17.5	12-21	19.5	12-27	0.28
Creatinine, mg/dL—mean ± SD	0.92	0.27	0.97	0.29	0.49
Hemoglobin, g/dL—mean ± SD	14.3	1.8	13.6	2.6	0.19
Platelet count, *10^3^/μL—mean ± SD	189	48	196	55	0.63
INR—median (IQR)	1.08	1.04-1.17	1.09	0.99-1.31	0.68
NTproBNP, pg/mL—median (IQR)	563	125-1577	284	93-1498	0.59
Pharmacotherapy—number (%)					
Prostacyclin analogue	42	100	0	0	n.a.
Endothelin receptor antagonist	21	50	17	45	0.66
Phosphodiesterase 5 inhibitor	40	95	34	90	0.42
Calcium channel blocker	4	10	8	21	0.21
Diuretic	18	43	20	53	0.50
β-blocker	12	29	14	37	0.47
Statin	22	53	19	50	1.00
Oral anticoagulant	7	17	7	18	1.00

Legend: ALT—alanine aminotransferase, BMI—body mass index, CVD—cardiovascular disease, INR—international normalized ratio, IQR—interquartile range, NTproBNP—N-terminal pro B-type natriuretic peptide, PAH—pulmonary arterial hypertension, SD—standard deviation, WHO—World Health Organization.

## Data Availability

Raw data, including flow cytometry data are available upon request.

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
