# Peer review of "Prostacyclin Analogues Inhibit Platelet Reactivity, Extracellular Vesicle Release and Thrombus Formation in Patients with Pulmonary Arterial Hypertension"

_jcm, 2021, doi:10.3390/jcm10051024_

Round 1
Reviewer 1 Report
This is a clinical study regarding the antiplatelet effect of prostacyclin analogues. The authors compared platelet function defined as platelet reactivity, platelet EVs concentration, and thrombus formation between PAH patients treated with or without parenteral prostacyclin analogues. In addition, the authors compared the antiplatelet effect of three different prostacyclin analogues. The authors concluded that PAH patients treated with prostacyclin analogues have impaired platelet function. However, the authors could not conclude the comparison of the antiplatelet effect of three different prostacyclin analogues because of incomplete results.
Major
- Patients’ number treated with iloprost was only 4 and only single case was tested EVs concentration and thrombus formation. The comparison between epoprostenol and treprostinil after excluding patients on iloprost would make the results and discussions much clearer. If the authors are still hoping to keep the patients in their cohort, they should indicate the results of EVs concentration and thrombus formation in at least additional one or two cases treated with iloprost.
- The authors indicated the correlation between the dosage of treprostinil and the plasma concentration of EVs (CD146) in Figure 6 whereas the plasma concentration of EVs (CD146) in patients treated with treprostinil was not significantly different from that of control group as indicated in Figure 3. It seems meaningless to seriously discuss about the correlation. In addition, the number of plots in Figure 6 were at least 25, much greater than patients’ number treated with treprostinil.
Minor
Table 2. Female gender -> Female gender – number (%)
Author Response
Dear Reviewer,
we are thankful for the time and effort that you spent to provide in-depth review of our manuscript. We corrected our manuscript according to your suggestions. Our response and corrections are listed below.
Major
Patients’ number treated with iloprost was only 4 and only single case was tested EVs concentration and thrombus formation. The comparison between epoprostenol and treprostinil after excluding patients on iloprost would make the results and discussions much clearer. If the authors are still hoping to keep the patients in their cohort, they should indicate the results of EVs concentration and thrombus formation in at least additional one or two cases treated with iloprost.
We agree that the comparison between various PGI2 analogues was incomplete due to a small number of patients in each group. Since the study was logistically demanding, with samples being collected in Poland and being measured in Netherlands and the device for thrombus formation being rent for the time of the study, it would not be easy to include additional patients on iloprost right now, which we regret. Nevertheless, we agree that excluding 1 patient on iloprost would make the comparison between epoprostenol and treprostinil more straightforward and we updated the figures by excluding iloprost (please see Figure 2,3,4).
The authors indicated the correlation between the dosage of treprostinil and the plasma concentration of EVs (CD146) in Figure 6 whereas the plasma concentration of EVs (CD146) in patients treated with treprostinil was not significantly different from that of control group as indicated in Figure 3. It seems meaningless to seriously discuss about the correlation. In addition, the number of plots in Figure 6 were at least 25, much greater than patients’ number treated with treprostinil.
We agree that this correlation is entirely hypothesis-generating and does not provide any solid evidence regarding the effect of treprostinil on endothelial EVs. Therefore, we removed the discussion regarding this correlation and Figure 6 from our manuscript.
Minor
Table 2. Female gender -> Female gender – number (%)
Thank you for noticing this mistake, we corrected it.
Altogether, we are grateful for the in-depth revision of our manuscript and we hope that it will be considered for publication in “Journal of Clinical Medicine”.
On behalf of all Authors,
Sincerely,
Aleksandra Gasecka
Reviewer 2 Report
This well-written account of a well-designed study highlights the influence of pharmaceutical treatment on platelet reactivity, extracellular vesicle release and thrombus formation. The methodology is appropriate and well-executed. The study in itself makes a significant contribution to the knowledge in the field and will be of interest to the journal’s readership.
However, several questions require clarification and/or need to be addressed to fill a gap in the manuscript.
Demographics:
- Have the patients been screened for the presence of a BMPR2 mutation? This is particularly important, as certain BMPR2 mutations result in the upregulation of platelet adhesion genes. Therefore, please provide the mutation status of the patients, and assess any potential influences of the mutation on your measurements and include this in your discussion. If the mutation status of the patients is unknown, please state this fact in the limitations of the study.
- Due to the higher prevalence, but lower severity of PAH in female patients, an assessment of sex-specific influences on your data is required. Please include this as a supplementary figure.
- Please further elaborate on the differences between the control and treatment groups, i.e. why was the control group treated with prostacyclins? Were they
- The NTproBNP count appears to be considerably higher in the study group, indicating right ventricular dilatation and impaired systolic function. Can you please elaborate on this apparent difference?
Figures and Tables:
- Table 2: Please add a description of the columns for the 2 columns under study group and control group.
- Please indicate the statistical analysis used in the Figure legends, and indicate what the bars represent, i.e. SE or SEM
- The bars used in the figures do not appear to represent your data well. Could you please provide a graph with the single data points highlighted
- Figure 4D: Please ensure consistency in spacing, i.e. n=20 versus n = 4
Main text:
- Please set your findings into context with the published literature on the inhibition of platelet activation by prostacyclin (For example: Yusuf et al., 2017, Scientific reports 7:5582)
- Please include your hypothesis of the mechanism of action for prostacyclin inhibiting the response of AA and ADP. For example through inactivation of G-proteins or an interaction with other cells?
- Please ensure consistency throughout the manuscript for the use of British vs. American English. An example: Line 52 – remodelling (BE); Line 152 - aetiology (BE), but Line 220: standardized (AE).
- Please ensure that the formatting is consistent throughout, i.e. the word “Figure” in Line 340 versus Line 353
- Please make sure that an article is used when necessary.
- Please correct the following minor spelling, or grammatical mistakes:
- Line 65: the pulmonary …cells
- Line 67: corelate to no correlate with
- Line 71: Guidelines of THE… society
- Line 91: ERA – Please remember to define all abbreviations at their first use.
- Line 106: THE European Society of …
- Line 130: patients’ flow diagram
- Line 182: the Vesicle Observation Centre
- Line 208: redundancy issue “diluted”
- Line 257: delete “a”
- Line 258: Altogether, spelling error
- Line 262: Fisher’s
- Line 282: more, not most
- 285: numbers
- Line 376: Data not shown.
- Line 418: systems
- Line 438: either use the singular “A thrombus” (with the article), or the plural “thrombi”
Author Response
Dear Reviewer,
we are thankful for the time and effort that you spent to provide in-depth review of our manuscript. We corrected our manuscript according to your suggestions. Our response and corrections are listed below.
This is a well-written account of a well-designed study highlights the influence of pharmaceutical treatment on platelet reactivity, extracellular vesicle release and thrombus formation. The methodology is appropriate and well-executed. The study in itself makes a significant contribution to the knowledge in the field and will be of interest to the journal’s readership.
Thank you for appreciating our study design and methodology.
However, several questions require clarification and/or need to be addressed to fill a gap in the manuscript.
Demographics:
- Have the patients been screened for the presence of a BMPR2 mutation? This is particularly important, as certain BMPR2 mutations result in the upregulation of platelet adhesion genes. Therefore, please provide the mutation status of the patients, and assess any potential influences of the mutation on your measurements and include this in your discussion. If the mutation status of the patients is unknown, please state this fact in the limitations of the study.
Thank you for raising this point. Unfortunately, screening for BMPR2 mutation was beyond the scope of our study, which we stated in the limitations as follows: Moreover, our patients were not screened for the presence of a bone morphogenetic protein receptor type II (BMPR2) mutation. Inactivating mutation in the BMPR2 gene in patients with PAH has been linked to vascular smooth muscle cell proliferation and upregulation of platelet adhesion. Therefore, it cannot be excluded that the mutation status has potential influences on the parameters of platelet function, evaluated in this study.
- Due to the higher prevalence, but lower severity of PAH in female patients, an assessment of sex-specific influences on your data is required. Please include this as a supplementary figure.
We are grateful for this remark. We conducted the additional analysis and added it to the Results, as follows: Due to the higher prevalence, but lower severity of PAH in female patients, we compared all measured platelet function parameters in female and male patients and did not find any gen-der-associated differences, except for higher concentrations of EVs from platelets (CD61+) and leukocytes (CD45+) in female (p≤ 0.03 for both). To further evaluate the effect of potential confounders on CD61+ and CD45+ EVs, we performed a multivariate regression analysis taking into account the concentrations of CD61+ and CD45+ EVs above median as dependent variable and (i) age, (ii) gender (female), (iii) PAH severity (WHO class) and (iv) therapy with PGI2 analogues as independent variables. We found that both female gender and therapy with PGI2 analogues were independently associated with both EV subtypes. We attached the figures and tables to the Supplementary Materials 2 (Figure S1, Table S1, S2).
- Please further elaborate on the differences between the control and treatment groups, i.e. why was the control group treated with prostacyclins?
Since 2015 ESC/ERS Guidelines for the diagnosis treatment of pulmonary hypertension recommend the combined therapy including ERA and/or PDE5i and PGI2 analogues in PAH patients (WHO functional class II-IV), such therapy was also used it our study. We acknowledge that the best setting would be to compare patients treated only with PGI2 analogues vs. those not receiving PGI2 analogues. Nevertheless, the single therapy in the study group vs. no therapy in the control group would be difficult from ethical reasons. We added this point to the limitations, as follows: Moreover, since combined therapy including ERA, PDE5i and PGI2 analogues is recommended in PAH patients in WHO functional class II-IV, such therapy was also used it our study, which did not allow to compare the effects of monotherapy with PGI2 analogues in the study group vs. no therapy in the control group.
- The NTproBNP count appears to be considerably higher in the study group, indicating right ventricular dilatation and impaired systolic function. Can you please elaborate on this apparent difference?
These differences are likely because – despite the lack of statistically significant differences – prostacyclins as parenteral drugs are recommended for patients with more advanced disease. In contrast, oral medications that were mainly prescribed in the control group are dedicated to treating patients in WHO class II and early class III. Therefore, we added this observation when discussing the non-randomised study design in the Limitation section, as follows: The main limitation of our study is the non-randomized study design. For example, despite the lack of statistically significant differences, the largest part of patients in the study group had WHO class III PAH (50%) and the largest part of patients in the control group had WHO class II PAH (55%). Concurrently, the NTproBNP level was considerably higher in the study group, indicating more advanced right ventricular dilatation and impaired systolic function. Therefore, despite the differences in the primary and secondary end-points between the groups, the results should be interpreted with caution.
Figures and Tables:
Table 2:
- Please add a description of the columns for the 2 columns under study group and control group.
We added both the column titles and added the unit to every row to make the subsequent parameters clear for the reader (highlighted in yellow).
- Please indicate the statistical analysis used in the Figure legends, and indicate what the bars represent, i.e. SE or SEM
We indicated the statistical analysis and bars to each Figure, as follows: Values compared using Mann–Whitney U test and showed as median and interquartile range or Values compared using Kruskal-Wallis test with Dunn’s correction for multiple comparisons and showed as median and interquartile range.
- The bars used in the figures do not appear to represent your data well. Could you please provide a graph with the single data points highlighted?
We changed all graphs from bar type to dot type.
- Figure 4D: Please ensure consistency in spacing, i.e. n=20 versus n = 4
We apologise for the inconsistency. We changed it.
Main text:
- Please set your findings into context with the published literature on the inhibition of platelet activation by prostacyclin (For example: Yusuf et al., 2017, Scientific reports 7:5582)
We discussed the previous reports on the antiplatelet effects of prostacyclin as follows: Preliminary evidence regarding the antiplatelet role of prostacyclin in animal models dates back to 30 years ago, when it was showed that prostacyclin inhibits platelet adhesion and thrombus formation on subendothelium [30] and prevents thrombus formation in the microcirculation [31]. More recently, prostacyclin was showed to reverses platelet stress fibre formation causing platelet aggregate instability in vitro, which was mimicked by the adenylyl cyclase activator forskolin and prevented by inhibitors of protein kinase A [32].
- Please include your hypothesis of the mechanism of action for prostacyclin inhibiting the response of AA and ADP. For example, through inactivation of G-proteins or an interaction with other cells?
We elaborated on the hypothetical explanation of prostacyclin inhibiting the response of AA and ADP as follows: In our study, treprostinil which binds to IP and DP receptors only, decreased platelet reactivity in response to AA and ADP. IP and DP receptors are coupled to Gs-protein that activates adenylyl cyclase and stimulates cyclic adenosine monophosphate (cAMP) formation. High concentration of cAMP makes platelets resistant to activation by any agonist [43]. Although we did not find any effect of treprostinil on EV release, other authors showed that treprostinil decreased the concentrations of platelet EVs in paediatric PAH, with no effect on leukocyte and endothelial EVs [37]. Likely, different populations and small sample sizes account for these differences.
- Please ensure consistency throughout the manuscript for the use of British vs. American English. An example: Line 52 – remodelling (BE); Line 152 - aetiology (BE), but Line 220: standardized (AE).
We apologise for the inconsistency. We corrected the spelling to British English.
- Please ensure that the formatting is consistent throughout, i.e. the word “Figure” in Line 340 versus Line 353
We corrected all names of Figures and Tables to starting with a capital letter.
- Please make sure that an article is used when necessary.
We double-checked the references; all articles are used when necessary.
- Please correct the following minor spelling, or grammatical mistakes:
- Line 65: the pulmonary …cells
- Line 67: corelate to no correlate with
- Line 71: Guidelines of THE… society
- Line 91: ERA – Please remember to define all abbreviations at their first use.
- Line 106: THE European Society of …
- Line 130: patients’ flow diagram
- Line 182: the Vesicle Observation Centre
- Line 208: redundancy issue “diluted”
- Line 257: delete “a”
- Line 258: Altogether, spelling error
- Line 262: Fisher’s
- Line 282: more, not most
- 285: numbers
- Line 376: Data not shown.
- Line 418: systems
- Line 438: either use the singular “A thrombus” (with the article), or the plural “thrombi”
Thank you for underlying these mistakes, we corrected them according to your suggestion.
Altogether, we are grateful for the in-depth revision of our manuscript and we hope that it will be considered for publication in “Journal of Clinical Medicine”.
On behalf of all Authors,
Sincerely,
Aleksandra Gasecka
Round 2
Reviewer 1 Report
The manuscript has revised adequately. This reviewer has no further comments.
Reviewer 2 Report
389 - replace with "females" or "female patients"